# Veterinary Diagnostic Approach of Common Virus Diseases in Adult Honeybees

**DOI:** 10.3390/vetsci7040159

**Published:** 2020-10-21

**Authors:** Julia Dittes, Heike Aupperle-Lellbach, Marc O. Schäfer, Christoph K. W. Mülling, Ilka U. Emmerich

**Affiliations:** 1Centre for Applied Training and Learning, Faculty of Veterinary Medicine, Leipzig University, An den Tierkliniken 19, 04103 Leipzig, Germany; 2Institute of Veterinary Anatomy, Histology and Embryology, Faculty of Veterinary Medicine, Leipzig University, An den Tierkliniken 43, 04103 Leipzig, Germany; c.muelling@vetmed.uni-leipzig.de; 3LABOKLIN GmbH & CO.KG, Labor für klinische Diagnostik, Steubenstraße 4, 97688 Bad Kissingen, Germany; aupperle@laboklin.com; 4Friedrich-Loeffler-Institut, Federal Research Institute for Animal Health, Südufer 10, 17493 Greifswald, Insel Riems, Germany; Marc.Schaefer@fli.de; 5VETIDATA, Institute of Pharmacology, Pharmacy and Toxicology, Faculty of Veterinary Medicine, Leipzig University, An den Tierkliniken 39, 04103 Leipzig, Germany; emmerich@vetmed.uni-leipzig.de

**Keywords:** honeybee veterinary medicine, acute bee paralysis, chronic bee paralysis, deformed wing virus, *varroa* infestation, nosemosis

## Abstract

Veterinarians are educated in prevention, diagnosis and treatment of diseases in various vertebrate species. As they are familiar with multifactorial health problems in single animals as well as in herd health management, their knowledge and skills can be beneficial for the beekeepers and honeybee health. However, in education and in practice, honeybees are not a common species for most veterinarians and the typical veterinary diagnostic methods such as blood sampling or auscultation are not applicable to the superorganism honeybee. Honeybee colonies may be affected by various biotic and abiotic factors. Among the infectious agents, RNA-viruses build the largest group, causing covert and overt infections in honeybee colonies which may lead to colony losses. Veterinarians could and should play a more substantial role in managing honeybee health—not limited to cases of notifiable diseases and official hygiene controls. This review discusses the veterinary diagnostic approach to adult bee examination with a special focus on diagnosis and differential diagnosis of the common virus diseases Acute Bee Paralysis Virus (ABPV)-Kashmir Bee Virus (KBV)-Israeli Acute Paralysis Virus (IAPV)-Complex, Chronic Bee Paralysis Virus (CBPV) and Deformed Wing Virus (DWV), as well as coinfections like *Varroa* spp. and *Nosema* spp.

## 1. Introduction

In contrast to human medicine, one of the most prominent features of veterinary medicine is that a veterinarian has to deal with a large variety of species. Whereas dogs, cats and horses are common patients in a veterinary practice, the honeybee (*Apis mellifera* L.) is only rarely in contact with veterinarians as a patient. Official veterinarians and diagnostic laboratories are responsible for the analysis of honey samples and for animal disease control in case of, e.g., American foulbrood, but the general disease prevention and control are done by the beekeepers. The majority of the establishments for veterinary education in Europe accredited by ESEVT (European System of Evaluation of Veterinary Training) offer elective or no training in honeybee veterinary medicine. Only in 24 of 68 surveyed schools honeybee veterinary medicine is part of the curriculum as a separate subject [1]. This is a very limited consideration in the curriculum and veterinary practice for a food-producing animal with such a tremendous importance for global agriculture [2].

However, in recent years, the honeybee has aroused increasing interest among veterinarians. While only 9 veterinarians held a specialization about honeybees in Germany in 2014 [3], five years later, there were 17 bee veterinarians or nearly twice as many experts in this field [4]. Honeybee diseases are on the rise [5] and at the same time, a decline of honeybee colonies and beekeepers in Europe can be seen [6]. More and more, veterinarians publish specialist literature to deal with honeybee diseases [7,8]. At the supranational level, the European Parliament in 2008 called for the Commission “to incorporate into its veterinary policy, research into, and actions to tackle bee disease” [9], and in April 2011, a European Union Reference Laboratory for Bee Health was designated to coordinate diagnostic methods, disease monitoring and expert training [10,11].

In honeybee colonies, a large number of individuals together form a superorganism. Both this unit as well as the individual bee have to be investigated carefully. General principles of herd health management and hygiene concepts are well-known to veterinarians and thus, can be applied to the honeybee. Nevertheless, dealing with diseases in honeybees is different from the normal approach in veterinary medicine. Usual examination techniques, known from mammals, are not applicable to insect species, although the general approach and procedures are similar. Food control is not possible as the feed is provided by the environment and not by the beekeeper or the veterinarian [7]. The honeybee is a food-producing animal and assuring the quality of honey is an additional objective for veterinarians and beekeepers.

A variety of biotic and abiotic factors have an impact on honeybee colonies. Belsky et al. presented a broad overview of these stressors: habitat and climate changes, weather, the density of apiaries and food resources, as well as transportation of colonies, are external aspects influencing the honeybee [12]. Equally, the bees depend on intrinsic factors such as genetics and queen longevity [13]. The intensive agriculture with monocultures decreases the plant diversity and thus, the food supply for the honeybees. Between mass flowering of, e.g., rapeseed (*Brassica napus*) and sunflower (*Helianthus annuus*), pollen harvest severely declines [12] and limits adequate nutrition. Such an insufficient protein diet weakens the bees in defending against pathogens [14]. Various bacteria, microsporidia, viruses and pests cause bee diseases, among which viruses have become more relevant during recent years [5].

This review presents basic information on selected virus diseases and the process of colony examination and diagnostics from the veterinary perspective, with a focus on adult honeybees.

## 2. Virus Diseases in Honeybees and Contributing Factors

Viruses, mostly positive single-strand RNA viruses, are the largest class of honeybee infecting pathogens [13]. Over 20 bee viruses have been identified to date, including the Acute bee paralysis virus (ABPV), Kashmir bee virus (KBV), Israeli acute paralysis virus (IAPV), forming the ABPV-KBV-IAPV complex, Black queen cell virus (BQCV), Chronic bee paralysis virus (CBPV), Deformed wing virus (DWV) and Sac brood virus (SBV). Detailed reviews about the most important ones can be found in the literature [15,16,17,18]. Information on symptoms and transmission routes of ABPV-KBV-IAPV, CBPV and DWV, which is relevant to the veterinary diagnostic approach, will be presented in this review. Table 1 provides a summary of taxonomy, symptoms, affected castes and main transmission routes.

### 2.1. Acute Bee Paralysis Virus—Kashmir Bee Virus—Israeli Acute Paralysis Virus—Complex

ABPV, KBV and IAPV are three similar icosahedral viruses of the family *Dicistroviridae*. During the last years, this complex has been reported in association with colony collapse disorder (CCD), a phenomenon of severe colony weakening without visible worker bee mortality [7,20,21]. ABPV and IAPV cause symptoms such as trembling, paralysis, inability to fly, darkening and loss of hair from thorax and abdomen, affecting mostly individual bees and not the whole colony [22]. Hou and Chejanovsky describe experimental symptoms of IAPV such as disorientation, shivering wings, crawling and progressive paralysis until death in infection experiments [20]. ABPV and KBV are a reason for a sharp decline in the adult bee population [22].

Transmission of these viruses is possible vertically as well as horizontally. Furthermore, the *Varroa* mite plays an important role as a mechanical vector [15]. It transfers the virus while feeding from hemolymph of the bees and increases the viral load in the colony [15].

### 2.2. Chronic Bee Paralysis Virus

CBPV is an unclassified RNA virus that causes two different syndromes in honeybees. Type A, the paralytic form, is characterized by trembling, disorientation and ataxia. Type B affected bees show black, hairless and greasy shining abdomens. Both can occur in the colony at the same time and lead to massive worker bee losses [23].

Comprehensive information on this virus can be found in our case report presenting an overt infection with CBPV in two colonies [24].

### 2.3. Deformed Wing Virus

DWV is a picorna-like virus, that is often involved in winter losses of honeybee colonies. Two main variants, A and B, can be differentiated [25,26,27,28]. In most cases, it can be detected in colonies as a covert infection (see definition of covert under Section 3 in this review) without causing clinical symptoms [15]. Transmission routes are vertical (via eggs and sperm) or horizontal through larval food, trophallaxis or cannibalism of pupae during hygienic behavior [18]. However, *Varroa destructor* plays the key role in transmission as well as virulence and pathology of the virus [15]. Obvious symptoms are seen in the colony only if DWV replicates in the mite before being transferred to the bees. Clinical signs are deformed wings, bloated, discolored, shortened abdomens, hypoplastic glands and pupal death [29]. Infected bees are not viable and die within less than 67 h after emergence [15].

## 3. Veterinary Diagnostic Approach

As mentioned above, honeybee health is often influenced by many different factors [7]. Without clinical findings, the appearance of a pathogen in a colony does not constitute a disease. For that reason, the terms overt and covert infections were introduced by de Miranda and Genersch to describe honeybee virus diseases [30]. The descriptive terms overt and covert are widely used in insect virology. Overt infections are characterized by obvious clinical findings related to the virus infection and a high virus production rate. Acute and chronic forms are differentiated. In covert infections, low titers of virus particles are present in the absence of clinical symptoms. Vertical transmission allows virus persistence over several generations and competent virus particles can turn into overt infections due to various influencing factors. Persistent infections with low-level virus production can be distinguished from latent infections without virus production [30].

From the veterinary perspective, a holistic diagnostic work-up of medical issues in honeybee colonies is important, because the environment, the colony, the pathogens and every bee are each just a link in the chain leading to occurrences of infections [7]. Figure 1 shows a detailed plan for diagnostics and management in general. Starting from the environmental observations, followed by an examination of the hive, the colony and the bees, samples are taken, and relevant laboratory diagnostics carried out and further illnesses investigated. The resulting problem list leads to a prognosis and a management plan. The main goal is the healthy colony formed by fit individuals.

### 3.1. Medical History, Appeareance of the Hive and Environment

Like in any other species, clinical examination starts with a detailed anamnesis. Gathering information from the beekeeper gives a first overview about the acute problem, which often is superficial and skewed by incorrectly interpreted data [31]. Asking detailed questions can help: When did the beekeeper first observe symptoms or changes in the colonies? How frequently are the hives inspected during the season? How is *Varroa* infestation monitored and controlled? Is the beekeeper migrating the colonies? Maybe the beekeeper can also offer information about weather, crops in the region around the apiary or the density of apiaries in the vicinity—information needed for epidemiological evaluation.

The location and the environment around the beehive have an immense relevance to the occurrence of health problems and should therefore be carefully observed. The weather, the food range, the landmarks and further factors influence the bees in a similar way as a stable or field does to cattle, for example. A complete hazard analysis is shown by Formato and Smulders [10]. Vitally important are accessible water, identifiable food sources, climate and wind [31]. The hive model, its material and the hive’s general condition should be noticed. Fecal spots as well as traces of predators and pests may be seen during external examination.

Figure 2A,B show two apiaries in different locations. The size of the colony and the offered space in the hive should be evaluated [7,31].

### 3.2. Examination of the Alighting Board and Observation of the Entrance Hole

First pathologic findings can be identified during observation of the alighting board and entrance hole. The mass of bees flying in and out—depending on temperature, weather, time of day and state of colony development—provides a first impression about the strength of a colony. Dead bees as well as bees and larvae with alterations of the phenotype, carried out of the hive, may be found in front of the entrance. Veterinarian and beekeeper should take their time to observe the entrance hole carefully (Figure 3A,B). The way the bees fly, the manner of landing and the behavior of guarding bees offer a lot of information, because unusual flying behavior may be a neurologic symptom of virus diseases. Furthermore, the ground in front of the hive should be investigated for waste, feces, dead bees and larval mummies [7]. These observations, together with an assessment of the environment and hive from the outside, enable a first evaluation of the colony condition [31].

### 3.3. Clinical Examination of the Superorganism “Honeybee Colony” and Observations of Living Bees

The “internal” clinical examination of the bee colony and of individual bees are the next steps. It is important to have a look at the adult bees of all castes (workers, drones and the queen), as well as the brood in different stages and the hive material. Honeybees are social insects that can survive within the colony as a superorganism only. The colony is functional if each single individual performs its tasks. In the following, we will focus on the clinical findings in adult bees.

Clinical examination of the bees and the frames should be performed together with the beekeeper. This way, it is possible to analyze the beekeeper´s operations while handling the bees [7]. General features to assess after opening the hive are the colony behavior, the odor and the colony strength [7,31]. Normally, a honeybee colony smells like wax. If vinegar or foul smell is perceived, there is likely a disease. The intercomb-spaces, where bees are visible, can be counted to give a first idea of colony size. This might be confirmed later by estimating the number of bees, brood cells and food, e.g., using the Liebefeld method [32,33]. The bees are carefully examined to find body deformities, wing abnormalities like V-wings or K-wings, changes in color and size, phoretic *Varroa* mites as well as behavioral, neurological, social or digestive symptoms. Signs of CBPV disease are hairless, black bees and/or neurological symptoms such as trembling, circling or paralysis [23,24]. Abnormal wing position and wing form are shown in case of CBPV or DWV. Phoretic *Varroa* indicate an intense infestation rate with the mite. Phenotype changes, e.g., crippled bees, high mortality and paralysis symptoms, can also be a sign of intoxication. Table 2 lists different phenotype changes in honeybees, which can be seen in a sample of dead bees as well, and their possible causes.

While examining the bees, the frames are assessed: the color and brightness of the wax correspond to the age of the combs. Old wax, having been used in several brood cycles, is much darker than newly produced wax. The frames themselves can be clean or show fecal spots in case of digestive problems [34]. Pollen and honey stores are evaluated to get an idea about the alimentation of the colony.

It is advisable to have a look at how easily the frames can be removed from the hive by the beekeeper. If they are not easy to remove, propolis foraging activity or insufficient surveillance by the beekeeper could be the reason [7].

For all symptoms, it is determined whether they affect a small number of individuals or numerous bees, which would indicate a severe problem. Furthermore, it is relevant which castes of bees show the symptoms.

During the colony investigation, the queen is sought. It is observed whether she shows any abnormalities and whether the symptoms are equal to those of the workers. The queen does not always have the same virus load and symptoms as her colony [35].

### 3.4. Taking Samples for Laboratory Analysis

To investigate the phenotype of bees and pathological changes in more detail and to perform a laboratory analysis, bee samples have to be taken. Most suitable are living symptomatic bees, euthanized. However, in case of increased forager mortality, freshly dead bees from the soil in front of the hive could be appropriate as well, but it has to be considered that there can be false-negative results in virus diagnostics because RNA is unstable in the environment. Different diagnostic methods require a specified number of bees, a fact that should be considered when taking the samples. To assess the size of a sample, the following reference value can be considered: 100 mL are equivalent to approximately 330 honeybees and 31 g. The required number of bees is caught and euthanized by freezing (about 15 min, −20 °C) [8]. Other ways of euthanizing are using 96% ethanol, carbon dioxide or sampling after asphyxiation with sulfur. The sample with dead bees is sent chilled in an air-permeable case to the laboratory. In general, it is advisable to contact the responsible laboratory for information on size and condition of the sample.

#### 3.4.1. Examination of Dead Bees

Dead bees are examined, and their phenotype is described. They can be sorted according to their size and symptoms: Bees can be smaller than normal and show shortened abdomens (Figure 4A) when infected with several viruses, e.g., Deformed Wing Virus (DWV). More indicative for DWV, associated with a severe *Varroa* infestation load, are crippled wings in freshly hatched honeybees (Figure 4B). Williams et al. describe a ranking in six categories, according to the severity of wing abnormities [36]. Hairless black abdomens are seen in case of a CBPV infection [24] or as a result of genetics, alimentation within honeydew flow period or for mechanical reasons. Robbery and fighting as well as maturing in foragers may result in breaking hairs and black abdomens, which, however, only affects individuals, whereas genetics, alimentation and CBPV affect the whole colony. If the abdomens of several bees seem to be bloated, pressure on the abdomen may lead to a light-brown fluid leaking from the gut. In such cases, the bee´s gastro-intestinal tract can be pulled out for further investigation. Often, an extended proboscis is seen (Figure 4A), which is a more unspecific sign and not necessarily evidence for an intoxication. A summary of phenotype changes is listed in Table 2 above.

#### 3.4.2. Examination of Debris

A plastic drawer is put under the hive to sample debris. Examination of debris provides significant information about the health, development status and strength of a honeybee colony. Building materials such as wax scales, cell lids and drops of propolis, as well as cell components like pollen, sugar and melicitose crystals or drops of diluted food [37], can be found (Figure 5). The respective amounts of these components are indicators for the strength of the colony.

Furthermore, traces of predators can be seen: Parts of bees are a sign of wasps or mice in the hive (Figure 5B). Mice leave 3–8 mm long feces on the drawer. Smaller, dark brown feces belong to the greater wax moth, *Galleria mellonella*, which can also be detected by its creamy-white to grey larvae [7].

Wood chips or straw from the feeding trough or condensation water can be found, too.

Debris examination is additionally used for *Varroa* infestation control. The mite can be seen with the naked eye. Their number, counted after the drawer was under the hive for a defined time, can be used to quantify the infestation [7].

The validity of the information gathered from debris depends on the weather, the season and the time the drawer was under the hive [37]. In relation to these conditions, it constitutes an important part of examination.

### 3.5. Laboratory Diagnosis

#### 3.5.1. PCR to Detect Viral Diseases

This step has to be adjusted to the clinical observations in each specified case. Honeybee viruses are mainly detected by Real-Time Reverse Transcription (RT)-Polymerase Chain Reaction (PCR). In the following, the methods are described for CBPV, ABPV and DWV in the way they are performed at the Friedrich-Loeffler-Institute, the Federal Research Institute for Animal Health. A sample of 50 symptomatic bees is an appropriate sample size to submit to the laboratory.

From each bee sample, ten bees are homogenized using the gentleMACS^TM^ Dissociator. Total RNA is purified from 150 µL of clarified bee homogenate using the RNeasy Mini Kit (Qiagen, Venlo, NL) according to the manufacturer’s instructions. For each of the viruses, a one-step real-time RT-PCR is subsequently performed in duplicate using the AgPathID^TM^ One-Step RT-PCR Kit (Applied Biosystems^TM,^, Waltham, MA, USA) in a 96-well reaction plate with 2.5 µL RNA in a final volume of 12.5 µL.

For CBPV detection, it contained 320 nM of forward and reverse primer (qCBPV 9: 5′- CGC AAG TAC GCC TTG ATA AAG AAC -3′; qCBPV 10: 5′- ACT ACT AGA AAC TCG TCG CTT CG -3′), and 200 nM of the CBPV probe (CBPV 2 probe: 5′- FAM- TCA AGA ACG AGA CCA CCG CCA AGT TC -BHQ1 -3′) [38].

For ABPV detection, it comprised 800 nM of forward and reverse primer (ABPV1: 5′- CAT ATT GGC GAG CCA CTA TG -3′; ABPVRn: 5′- CTA CCA GGT TCA AAG AAA ATT TC -3′), and 90.4 nM of the ABPV probe (ABPVnTaq: 5′- FAM- ATA GTT AAA ACA GCT TTT CAC ACT GG -BHQ1 -3′) [39].

For DWV-A detection, it contained 350 nM of forward and reverse primer (F-DWV_4250: 5′- GCG GCT AAG ATT GTA AAT TG -3′; R-DWV_4321: 5′- GTG ACT AGC ATA ACC ATG ATT A -3′), and 100 nM of the DWV-A probe (Pr-DWV_4293: 5′- FAM- CCT TGA CCA GTA GAC ACA GCA TC -BHQ1 -3′) [40]. For DWV-B detection, it contained 1.2 µM of forward and reverse primer (F-VDV1_4218: 5′ GGT CTG AAG CGA AAA TAG -3′; R-VDV1_4290: 5′- CTA GCA TAT CCA TGA TTA TAA AC -3′), and 400 nM of the DWV-B probe (Pr-VDV1_4266: 5′- FAM- CCT TGT CCA GTA GAT ACA GCA TCA CA -BHQ1 -3′) [40].

The thermal cycling conditions are 10 min at 45 °C (RT = reverse transcription), 10 min at 95 °C (RT inactivation, initial denaturation, activation of DNA Polymerase), followed by 41 amplification cycles at 94 °C for 15 s and 60 °C for 45 s. The results are expressed as the mean of the two replicates for each reaction. Figure 6 shows an amplification diagram for evaluation. The assay is combined with an internal control assay in a duplex real-time RT-PCR—with the exception of DWV-B detection. This is carried out as a simplex approach, since another primer–probe combination interferes with the PCR approach and hinders or prevents the amplification of the virus fragment. DWV-B therefore is detected in parallel to the closely related DWV-A virus, which also helps in differentiating DWV disease. The performed internal controls are, on the one hand, a universal internal control system based on IC2-RNA [41], and, on the other hand, the detection of β-actin in the extracted samples [42].

Both internal controls monitor that the RNA extraction was successful in all samples (and extraction controls), as well as confirm their transcription into cDNA and the amplification of those during real-time RT-PCR. Furthermore, this allows to see whether a uniform effective RNA extraction took place.

#### 3.5.2. Monitoring the *Varroa* infestation

The *Varroa* mite is one of the main stressors, probably globally “the greatest threat” [7] for honeybee colonies. If there is no control, colonies with a high mite load are weakening until they ultimately collapse. Additionally, *Varroa destructor* serves as a mechanical and biological vector for various honeybee viruses and suppresses the immune response of honeybees [43]. Therefore, it should be monitored carefully.

To evaluate the mite load in a colony, different methods are described in the literature. It is possible to examine the debris, the brood and adult bees. Drone brood can be investigated by removing the lids of the cells and studying the larvae and combs individually (Figure 7A). In case of a larger number of samples, the brood cells can be washed out into a sieve system where mites and brood are gathered separated from each other [44].

Adult bees play a role in assessing the mite load (Figure 7B). Examination of the natural mite fall in the debris is a common method to determine the mite load. The natural mite fall is the number of mites per day falling naturally from the bees down to the bottom board. A drawer is placed under the hive over a period of 2 to 5 days (Figure 8A). It can be lined with a sticky paper towel. The drawer is examined, and the mites are counted (Figure 8B). The oval, reddish-brown *Varroa* mites are 1.2 to 1.7 mm [7] and can easily be detected in the debris (Figure 8C). Finally, the number of mites per day has to be calculated.

Bak et al. compared the flotation and the powdered sugar shake method in 2009 with similar results for both [45]. About 300 young adult bees (is equivalent to 100 mL) are sampled from a frame with uncapped brood for either flotation or powdered sugar method. For flotation, the bees are shaken in alcohol solution for five minutes, after freezing or killing them in alcohol. The mites are removed from the bees and can be counted (Figure 9).

The second method leaves the bees alive. Dusted with icing sugar, they are shaken gently two times for two minutes. The grooming behavior of the bees is stimulated and the mites’ feet do not stick any longer, thus, the mites are dislodged from the bees (Figure 10). Either the sugar is sieved out or dissolved in water, so that the separated mites can be counted [45,46]. If the mite infestation rate is under 5%, the colony is slightly affected, if the value is above 10%, an immediate treatment should be applied [7].

#### 3.5.3. Detection of Nosemosis

Fecal spots at or in honeybee hives can be a sign for nosemosis, but they do not have to be [34]. Dysentery can be caused by different pathogens such as CBPV, *Malpighamoeba mellificae* and *Nosema*-species, or be a symptom of stress. In 21 examined honeybee colonies with fecal spots, only 43% could be proven to be *Nosema*-positive [34].

*Nosema*-species can occur in a honeybee colony as a covert infection without harming it. But, it can serve as an additional and weakening stressor to the honeybees. In some cases, it can be a precursor of other diseases by opening infection routes due to epithelial damage in the midgut [47].

For investigation of nosemosis, abdomens of 20 freshly dead bees are separated in a mortar (Figure 11A). They are crushed with the pestle while adding 5 mL of aqua purificata (Figure 11B). Then, water is added to form a solution of 1 mL per bee. A drop of that sample is put onto a microscope slide and covered with a coverslip without trapping air bubbles (Figure 11C). The slide is examined under the microscope at 400× magnification [48]. This solution is suitable for detection of both *Nosema* and *Malpighamoeba mellificae*.

*Nosema* spores are oval-shaped and about 4 to 7 µm in length and 2 to 4 µm in width. The two species, *N. apis* and *N. ceranae,* cannot be distinguished by this method [7]. For further differentiation, a PCR would be required [49].

An estimation of the number of spores can be carried out using the scheme described by Ritter in 1996 [50]. The spores seen per visual field are counted and classified into three categories: Less than 20 spores per visual field signify a slight infestation, between 20 and 100 spores per visual field indicate a moderate infestation and more than 100 spores constitute a severe infestation [50]. The three grades are presented in Figure 12. For a more specific result, a hemacytometer should be used [48].

#### 3.5.4. Intoxications

If a massive number of dead bees is found in front of a hive or crippled bees are seen in the colonies, the beekeeper will often suspect a bee poisoning incident with a bee damage as a result of exposure to toxic plant protection products. However, there are some facts which have to be considered. Are all colonies in an apiary affected by increased mortality or is it only one or two? Is the apiary near fields making an application of a pesticide possible? And, are there other symptoms occurring?

In Germany, the Institute for Bee Protection in the Federal Research Centre for Cultivated Plants, Julius-Kühn-Institut (JKI), in cooperation with the national plant protection services examines samples of honeybees with suspected poisoning and plants. The bee sample undergoes a pollen analysis with pollen from the pollen basket (corbicula) or bee hairs to narrow down the incident to plants, which the bees visited. A *Nosema*-detection is performed, because *Nosema*-infected bees are more susceptible to intoxications. Finally, a bioassay with larvae of *Aedes aegyptii* L. is done to detect the presence of bee toxic insecticides. In 2018, 141 samples were sent in with the suspicion of bee intoxication, but only 61 samples were suitable for investigation. In 19 of these 61 incidents, bee toxic insecticides have been detected [51]. Further information can be found on the JKI website [52].

The numbers clearly illustrate the relevance of sending in appropriate bee material. An appropriate sample for this investigation should consist of 1000 bees (is equivalent to 100 g). The sample should be taken under witness during the first 24 h after seeing symptoms and immediately sent chilled to the investigating institution [53]. The basic veterinary knowledge about correct sampling can be valuable here for taking an appropriate bee sample and, if possible, a suspicious plant sample as well as taking and storing retention samples in case of subsequent questions. A detailed photographic documentation of the incident is advisable.

### 3.6. List of Medical Issues, Diagnoses and Prognosis

After examinations, the next step is to list medical issues and establish a final diagnosis. According to the list of medical issues, a prognosis is derived, and a treatment plan created. An example in case of an overt CBPV infection can be found in Dittes et al. [24]. Investigations show that asymptomatic healthy colonies may have higher virus loads than diseased colonies [54,55]. Therefore, the laboratory results must precisely be linked to the information gathered by anamnesis, clinical findings and observations to evaluate the relevance of pathogens. The results form the basis for the choice of therapeutic measures.

If a notifiable disease is diagnosed, the relevant authorities have to be informed. There are notifiable diseases to the World Organization for Animal Health (OiE) and to the veterinary authorities in each country, depending on local regulations. The Terrestrial Animal Health Code lists infections with *Paenibacillus larvae* and *Melissococcus plutonius* and infestations with *Acarapis woodi*, *Varroa* spp., *Aethina tumida* and *Tropilaelaps* spp. in Article 1.3.8 [56]. In Germany, infections with *Paenibacillus larvae* and infestations with *Aethina tumida* and *Tropilaelaps* spp. are notifiable. Currently, virus diseases form a part to the notifiable diseases only in Romania [7].

### 3.7. Outcome Control

An infected honeybee colony should be monitored closely during disease management to recognize problems and success as soon as possible and intervene in case of complications. It is advisable to use a logbook to record detailed information on every colony during disease management as well as during the whole season. Veterinarians are familiar with the management and interpretation of patient records as well as drawing conclusions out of them. A detailed logbook is important to gain basic information to choose and apply appropriate therapy concepts for disease control. The main point to prevent DWV infection and disease, as well as further *Varroa*-associated diseases, is to control the *Varroa* mite infestation. Further studies have to show recommendable therapy measures to control other viral diseases e.g. CBPV in honeybee colonies with overt infections.

## 4. Conclusions

A holistic approach to bee disease diagnosis is important to establish a correct and comprehensive diagnosis and save colonies. Disease outbreaks in honeybees are often tied to more complex interactions than in other species. Especially, the biological form of life in a superorganism has to be considered during the entire course of examination and management. Furthermore, the diagnostic techniques and possible ways of treatment are more limited than in the “normal” veterinary patient. There are no vaccinations and less veterinary medical products available for honeybees. The lifetime of an individual bee and the dependence on the season have to be considered.

A detailed and careful investigation is the base of diagnosis, and treatment decisions are to be made focusing on the colony as a whole rather than the individual animal. Depending on the colony location, the number of colonies in an apiary and beekeeping strategies as well as further influencing factors, diagnostics and the management plan have to be adapted to the individual requirements.

Veterinarians can be a valuable asset to the beekeeper because of their ability to prevent, diagnose and treat diseases in various species and in populations. In the future, veterinarians should recognize the important role and also the opportunities they can have within the honeybee sector. Taking the chance offered, the veterinarians may be part of education of beekeepers, part of sanitary audits and part of bee health overall in the veterinary practice, in addition to their official function in Veterinary Authorities.

## Figures and Tables

**Figure 1 vetsci-07-00159-f001:**
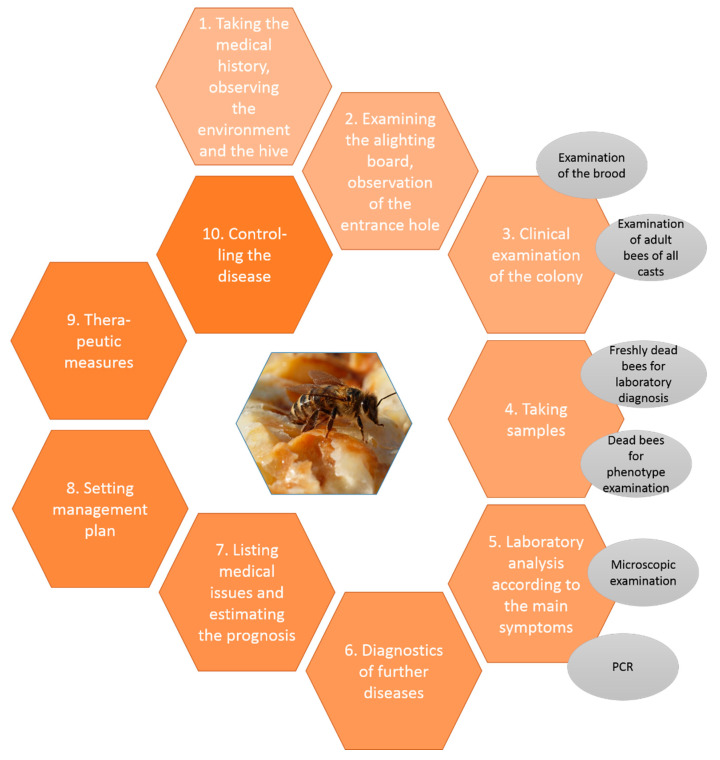
Diagnostic approach in bee diseases, © Julia Dittes.

**Figure 2 vetsci-07-00159-f002:**
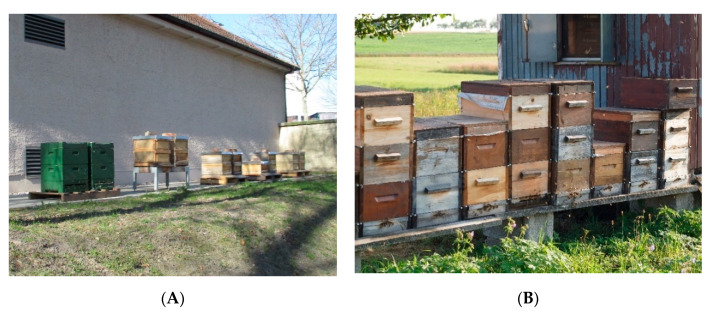
(**A**) Apiary in April 2019: colonies with two brood chambers in wooden Zander hives and polystyrene Segeberger Classic hives, standing in pairs of two. © Julia Dittes. (**B**) Apiary with Zander hives with one or two brood chambers and honey supers on top, standing next to each other. © Heike Aupperle-Lellbach.

**Figure 3 vetsci-07-00159-f003:**
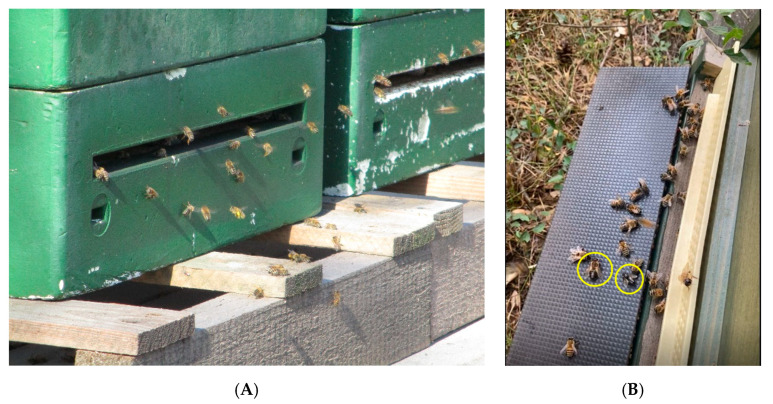
(**A**) Observation of the entrance hole enables a first evaluation of colony strength and activity according to season, weather and environment. © Julia Dittes. (**B**) Alighting board of a Chronic Bee Paralysis Virus-infected honeybee colony with some black hairless individuals (yellow circles). © Silvia Heisch.

**Figure 4 vetsci-07-00159-f004:**
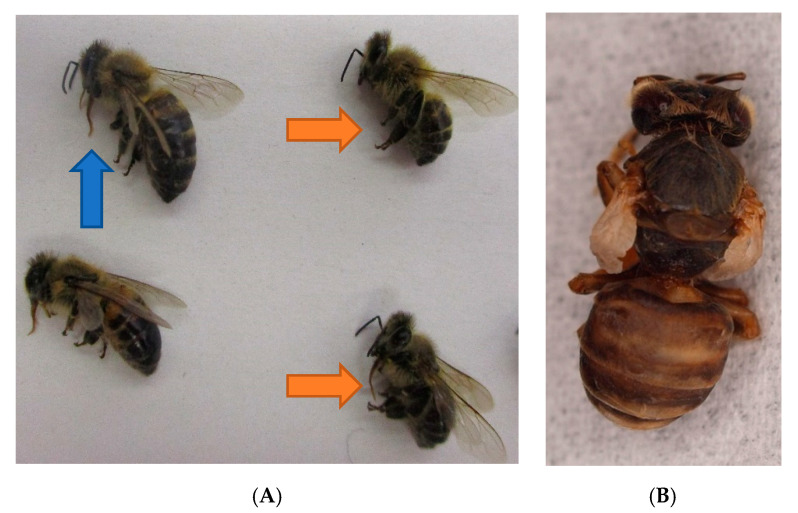
(**A**) sample of dead bees of a honeybee colony infected with CBPV in 2019: inhomogeneous size of worker bees, shortened abdomen (orange arrows), extended proboscis (blue arrow); © Julia Dittes. (**B**) freshly hatched honeybee infected by DWV; © Heike Aupperle-Lellbach.

**Figure 5 vetsci-07-00159-f005:**
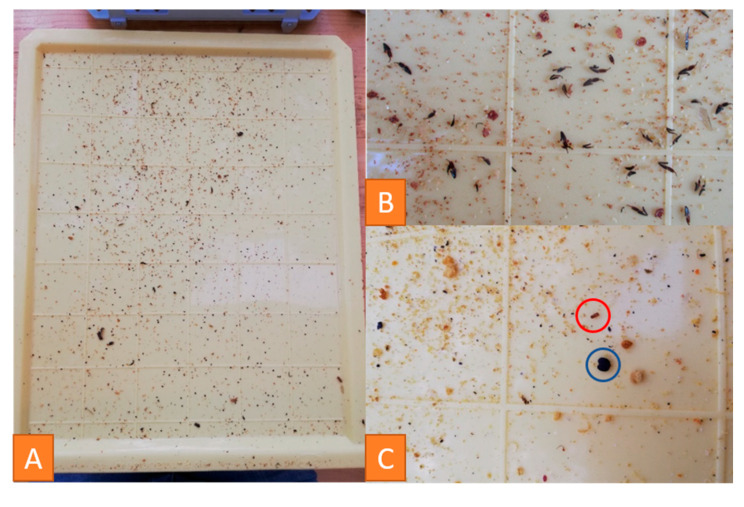
Photographs of a plastic drawer after three days under the hive of an *Apis mellifera carnica* colony in September with debris containing a lot of *Varroa* mites (**A**), separated legs and wings of bees (**B**), pollen (blue cycle) and wax cylinders (red cycle) (**C**). © Julia Dittes.

**Figure 6 vetsci-07-00159-f006:**
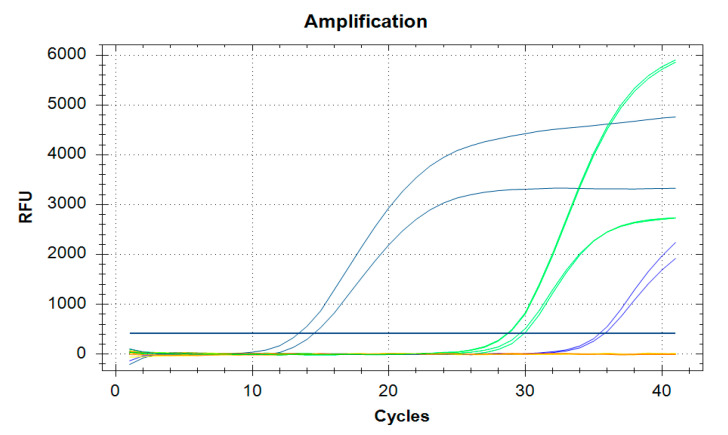
Amplification of a sample of a honeybee colony investigated for CBPV and ABPV in duplicate, result: ABPV = Ct 35.23, CBPV = Ct 14.33; green = positive controls, blue = CBPV, violet = ABPV, where the sample lines cross the blue straight line, Ct is determined, © Julia Dittes.

**Figure 7 vetsci-07-00159-f007:**
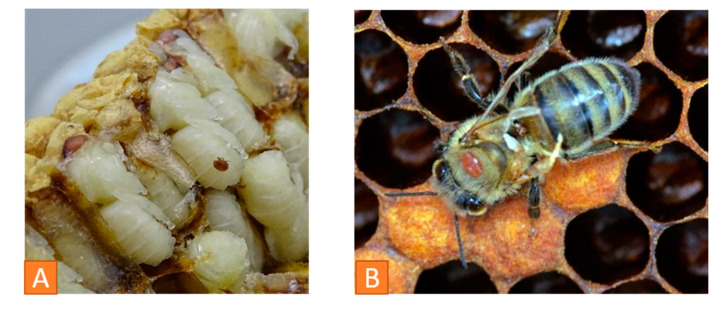
(**A**) *Varroa* infestation control via examination of drone brood: red-eyed larva with *Varroa* mite, © Julia Dittes. (**B**) DWV-infected adult honeybee with phoretic *Varroa* mites on thorax, © Ilka Emmerich.

**Figure 8 vetsci-07-00159-f008:**
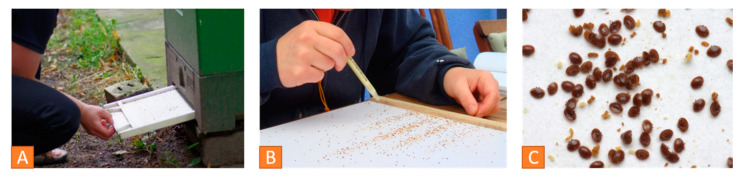
*Varroa* infestation control via debris examination: (**A**) A drawer with a sticky paper is put under the hive, (**B**) counting the number of *Varroa* mites, (**C**) close-up of *Varroa mites* on the drawer. © Jens Emmerich.

**Figure 9 vetsci-07-00159-f009:**
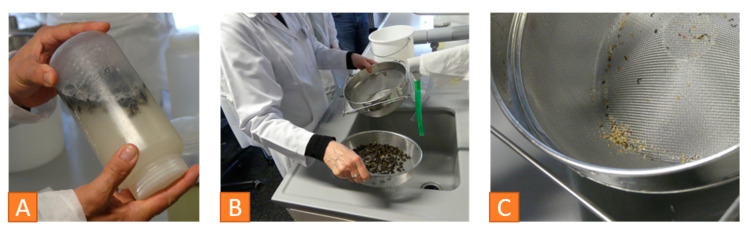
*Varroa* infestation control with the alcohol washing method, (**A**) shaking the honeybee sample in the ethanol solution, (**B**) straining the liquid in a sieve system, (**C**) mites in the smaller sieve can be counted. © Heike Aupperle-Lellbach.

**Figure 10 vetsci-07-00159-f010:**
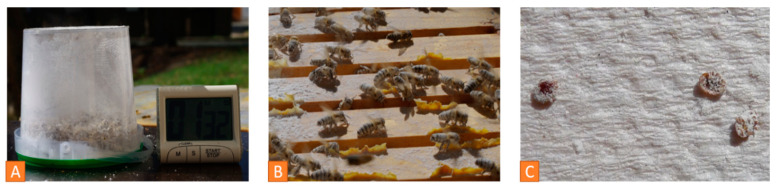
*Varroa* infestation control via iced sugar shaking method, (**A**) iced sugar-dusted bees in a shaking bucket, (**B**) dusted bees back in the hive after the procedure, (**C**) dusted mites on a paper towel. © Ilka Emmerich.

**Figure 11 vetsci-07-00159-f011:**
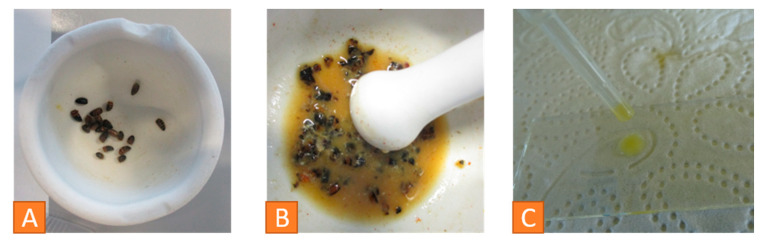
Detection of Nosemosis: (**A**) abdomens of 20 honeybees in a mortar, (**B**) crushing the abdomens with aqua purificata, (**C**) drop of the sample on a microscopic slide. © Julia Dittes.

**Figure 12 vetsci-07-00159-f012:**
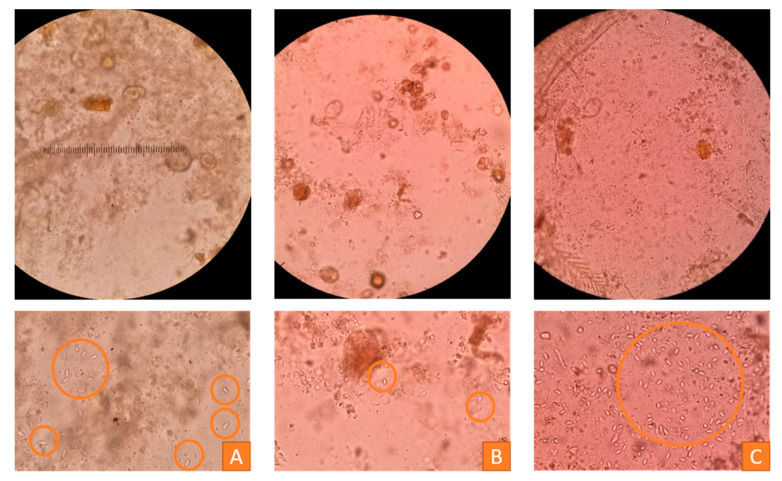
Microscopic investigation of bee samples of two honeybee colonies to estimate nosemosis infestation (400×), orange circles = Nosema spores (oval-shaped, 4–7 µm in length, 2–4 µm in width), upper row = visual fields, lower row = close-ups. (**A**) Sample of a colony with a moderate infestation (between 20 and 100 spores per visual field), (**B**) sample of a colony with a slight infestation (less than 20 spores per visual field), (**C**) sample of a colony with a severe infestation (more than 100 spores per visual field), © Julia Dittes.

**Table 1 vetsci-07-00159-t001:** Overview of selected honeybee viruses (taxonomy, symptoms, affected casts of bees and transmission routes), modified after Vidal-Naquet [19].

Virus	Taxonomy	Symptoms in a Colony	Affected Castes	Transmission (Main Routes)
Chronic bee paralysis virus (CBPV)	unclassified (RNA)	hairless black syndrome and paralysis syndrome, high mortality after a few days	w, d, q	c, o
Acute bee paralysis virus (ABPV)	*Dicistroviridae* (RNA)	paralysis and high mortality after 1–2 days (experimentally)	w, d	vec, ver (v, to)
Kashmir bee virus (KBV)	*Dicistroviridae* (RNA)	mortality without other symptoms	w, d	vec, ver
Israeli acute paralysis virus (IAPV)	*Dicistroviridae* (RNA)	paralysis and death	w, d	vec, ver
Deformed Wing Virus (DWV)	Picorna-like virus (RNA)	crippled bees with deformed wings and shortened abdomens	w, d, q	vec, ver (v, to)

Legend: RNA = Ribonucleic acid, w = worker, d = drone, q = queen, c = direct contact, o = oral–fecal, vec = vector-borne, ver = vertical, v = venereal, to = transovarial.

**Table 2 vetsci-07-00159-t002:** Phenotype changes in honeybees and their possible causes.

Phenotype Feature	Possible Causes (Selection)
shortened abdomens	DWV, Varroosis, CBPV
hairless, black abdomens	CBPV, black robbers for alimentary, genetic or mechanical reasons
crippled wings, legs, antennae	Varroosis, DWV, intoxication, CBPV
bloated abdomens and diarrhea (pressure on abdomen light-brown fluid)	Nosemosis, *Malpighamoeba mellificae,* CBPV
extended proboscis	virus diseases, intoxication, unspecific

Legend: DWV = Deformed wing virus, CBPV = Chronic bee paralysis virus.

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
