# Peer review of "Veterinary Diagnostic Approach of Common Virus Diseases in Adult Honeybees"

_vetsci, 2020, doi:10.3390/vetsci7040159_

Round 1

Reviewer 1 Report

Basically, this manuscript is well written and copuld be very usefull for veterinaries that start working with honeybees. This situation, that veterinarians are not prepare to work with honeybees, is comon around the world. So, this manuscript would be really usefull for them. 

This version clearly improve the original version. 

However, there is an important mistake that must be correct. In line 220-221 it is written "However, in case of increased forager mortality, dead bees from the soil in front of the hive are more appropriate". This type of samples could be use only to check same samples, but for the case of honeybee viruses this samples are not usefull, because the RNA is unstable and degardes at room temperature. This sentence must be correct.

Author Response

Manuscript vetsci-955750: Veterinary diagnostic approach of common virus diseases in adult honey bees

The authors thank the reviewers for their thoughtful comments on the manuscript. We have attempted to address all comments and believe that the manuscript has been improved markedly.

However, there is an important mistake that must be correct. In line 220-221 it is written "However, in case of increased forager mortality, dead bees from the soil in front of the hive are more appropriate". This type of samples could be use only to check same samples, but for the case of honeybee viruses this samples are not usefull, because the RNA is unstable and degardes at room temperature. This sentence must be correct.

Reply:

Thank you for your comment. We changed the sentence into the following:

  1. 221-223: „However, in case of increased forager mortality, freshly dead bees from the soil in front of the hive could be appropriate as well, but it has to be considered, that there can be false-negative results in virus diagnostics because RNA is unstable in the environment.”

Reviewer 2 Report

This is an interesting digest of the current status of disease diagnosis in honey bees. I appreciate the authors' input and the fact that the article encompasses most of our current understanding of the subject. In addition, the authors have taken the approach of emphasizing that the subject should be under the purview of trained veterinarians, which I agree with. I do not see any need to comment about the structure or the presentation of the manuscript. I welcome this review.

Minor issues:

210 "beekeeper could be the reason"

391 Suggest that the term "intoxication" be replaced with some form of phrase suggesting xeno-chemical toxicity. Intoxication can be misleading suggesting honey fermentation, although in the strictest term authors are not wrong.

431 Section "Outcome Control" could be expanded depending on the Journal's acceptable limitations on the number of pages. Would a veterinarian approach the issue(s) differently than the current techniques involving disposal of the hive or use of oxalic acid or menthol, etc. for varroa and beetles? Or do the authors see any upcoming means of controlling viral diseases in a colony of numerous infected individuals?

Author Response

Manuscript vetsci-955750: Veterinary diagnostic approach of common virus diseases in adult honey bees

The authors thank the reviewers for their thoughtful comments on the manuscript. We have attempted to address all comments and believe that the manuscript has been improved markedly.

Minor issues:

  1. 210 "beekeeper could be the reason"

Reply:

Thank you for your comment. I changed the sentence as required. See line 211.

391 Suggest that the term "intoxication" be replaced with some form of phrase suggesting xeno-chemical toxicity. Intoxication can be misleading suggesting honey fermentation, although in the strictest term authors are not wrong.

Reply:

Thank you for your comment. The term intoxication now is described with the phrase bee poisoning incident as it is used at the English website of Julius-Kühn-Institut.

lines 393-394: “…, the beekeeper will often suspect a bee poisoning incident with a bee damage as a result of exposure to toxic plant protection products”

431 Section "Outcome Control" could be expanded depending on the Journal's acceptable limitations on the number of pages. Would a veterinarian approach the issue(s) differently than the current techniques involving disposal of the hive or use of oxalic acid or menthol, etc. for varroa and beetles? Or do the authors see any upcoming means of controlling viral diseases in a colony of numerous infected individuals?

Reply:

Thank you for your comment. We added some sentences in this part, as you can see in lines 432-436. Therapy measures were not meant to be part of this article. At the moment we are doing some studies about therapy measurements in case of CBPV overt infections and hope to draw conclusions about recommendable concepts to control viral diseases. Important for that is detailed information recorded by beekeepers and veterinarians in the beekeeping sector.

lines 434-438: “Veterinarians are used to the management and interpretation of patient records as well as drawing conclusions out of them. A detailed logbook is important to gain basic information to choose and apply appropriate therapy concepts for disease control. Further studies have to show recommendable therapy measures to control viral diseases in honey bee colonies with overt infections.”